# GANDALF: Gated Adaptive Network for Deep Automated Learning of Features for Tabular Data

## Abstract

We propose a novel high-performance, interpretable, and parameter & computationally efficient deep learning architecture for tabular data, Gated Adaptive Network for Deep Automated Learning of Features (GANDALF). GANDALF relies on a new tabular processing unit with a gating mechanism and in-built feature selection called Gated Feature Learning Unit (GFLU) as a feature representation learning unit. We demonstrate that GANDALF outperforms or stays at-par with SOTA approaches like XGBoost, SAINT, FT-Transformers, etc. by experiments on multiple established public benchmarks. We have made available the code at `https://anonymous.4open.science/r/gandalf-43BD` under MIT License.

## 1 Introduction

Deep Learning (Goodfellow et al., 2016) has revolutionized the field of machine learning, surpassing state-of-the-art systems in various domains such as computer vision (Krizhevsky et al., 2017), natural language processing (Devlin et al., 2019), reinforcement learning (Silver et al., 2016). However, it has not made significant strides in tabular data analysis where state-of-the-art techniques typically rely on *shallow* models such as gradient boosted decision trees (GBDT) (Friedman, 2001; Chen & Guestrin, 2016; Ke et al., 2017; Prokhorenkova et al., 2018).

Recognizing the importance of deep learning in this context, several new architectures have been proposed(Arik & Pfister, 2021; Popov et al., 2020) and a PyTorch-based deep learning library has been developed (PyTorch Tabular (Joseph, 2021)). Surveys conducted by Shwartz-Ziv & Armon (2022) and Borisov et al. (2021) highlight the existing gap between deep learning models and the GBDT state-of-the-art in terms of performance, training, and inference times. This discrepancy is further substantiated by the prevalence of shallow GBDT models in Kaggle competitions.

In this paper, our aim is to narrow this gap by proposing novel approaches and techniques. We introduce *Gated Adaptive Network for Deep Automated Learning of Features (GANDALF)*, a new deep learning architecture for tabular data. In the following sections we will explain the design choices which make GANDALF a competitive model choice for tabular data. Through a large number of experiments on public benchmarks, we compare the proposed approach to leading GBDT implementations and other deep learning models and show that our model is accurate, parameter efficient, fast, and robust with lesser number of hyper parameters to tune.

Overall, our main contributions can be summarized as follows:

1. A new tabular data processing unit, the Gated Feature Learning Unit (GFLU), which can be used in place of regular Multi Layer Perceptron (Amari, 1967). The GFLU has feature selection and interpretability built into its design, making it a powerful and flexible tool for tabular data analysis.

2. A new gating mechanism for tabular data, successfully adapting Gated Recurrent Units (Cho et al., 2014) to non-temporal setting. This mechanism allows to learn more complex and informative representations of tabular data, which leads to improved performance on downstream tasks. To the

best of our knowledge, this is the first time such an approach has been used to process non-temporal tabular data.

3. Evaluation with public benchmarks instead of handpicked datasets to show the superiority of the proposed approach.

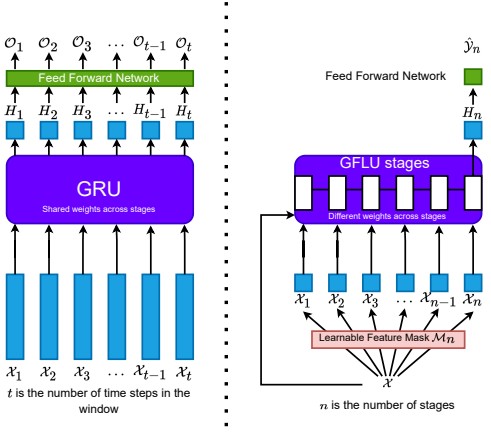

Figure 1: GRU vs GFLU - The schematic shows how the two units of learning are different - namely the different weights per stage, the learnable feature mask at each stage, and using only the last hidden state for further processing.

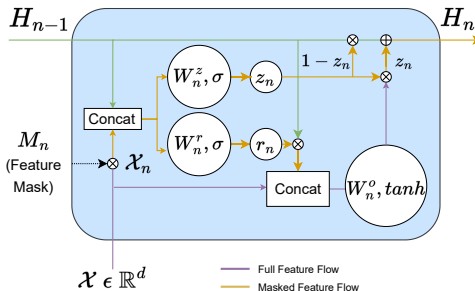

Figure 2: Detailed View of the Gated Feature Learning Unit. $\otimes$ represents element wise multiplication and $\oplus$ addition

## 2  Related Work

**Deep Learning approaches for tabular data:** While gradient boosted decision trees (GBDTs) remain the state-of-the-art approach for tabular data analysis, the deep learning community has made significant progress in this domain. Models such as TabNet (Arik & Pfister, 2021), Neural Oblivious Decision Trees(NODE) (Popov et al., 2020), FT-Transformer (Gorishniy et al., 2021), Net-DNF (Katzir et al., 2021) and DANet (Chen et al., 2021) have demonstrated competitive performance, often outperforming or matching the popular "shallow" GBDT models. NODE (Popov et al., 2020) combined neural oblivious decision trees with dense connections and achieved comparable performance to GBDTs. Net-DNF (Katzir et al., 2021) employed soft versions of logical boolean formulas to aggregate results from numerous shallow fully-connected models. NODE and Net-DNF followed ensemble learning approaches, utilizing a large number of shallow networks, resulting in high computational complexity. TabNet (Arik & Pfister, 2021) sequentially computed sparse attentions to mimic the feature splitting procedure of tree models. But TabNet has been shown to have inferior performance by Katzir et al. (2021) and Borisov et al. (2022). GANDALF, on the other hand, is much lower on the computational complexity and shows promising performance on public benchmarks for tabular data.

**Feature Representation Learning:**  Tabular data has well-defined features, but data scientists still spend a lot of time curating and creating new features. This shows the need for representation learning that can automate feature selection and feature interactions. Classical decision trees use information metrics, such as Gini index (Breiman et al., 1984), for feature selection while fully connected networks consider all the features and their interactions. This makes Decision Trees less prone to the noise. TabNet (Arik & Pfister, 2021) uses an attention mechanism for feature selection, but it operates at the instance level. NODE (Popov et al., 2020) uses learnable feature masks with the $\alpha$-entmax function (Peters et al., 2019) for feature selection within the context of the oblivious trees. DANet (Chen et al., 2021) also uses the $\alpha$-entmax function as feature masks and uses batch normalization along with learnable weights in a simple attention mechanism to learn representations. In this work, we use a similar approach to DANet but avoid the need for batch normalization and introduce learnable sparsity in the proposed novel gating mechanism.

**Gating Mechanism & Feature Selection:** Gating mechanisms and other Feature Selection techniques have gained significant popularity in deep learning for their ability to control the information flow within a network. These mechanisms have been widely applied across various domains, including computer vision, natural language processing (NLP), recommendation systems, and even tabular data analysis. Yamada et al. (2018) proposed using a Bernoulli Distribution for each feature and implementing a hard feature selection. Borisov et al. (2019) proposed a simpler soft feature selection using a learnable parameter and sigmoid activation. Liao et al. (2020) proposed a masking strategy using a softmax activation of a projection of the input features. GANDALF uses a soft feature selection, but using sparse activation functions to encourage sparse masks, similar to the feature masks in Arik & Pfister (2021). In sequence models like LSTM (Hochreiter & Schmidhuber, 1997), GRU (Cho et al., 2014), and Highway Networks (Srivastava et al., 2015), gating mechanisms are commonly employed to mitigate issues like vanishing and exploding gradients. Another notable component that incorporates gating is Gated Linear Units (Dauphin et al., 2017; Gehring et al., 2017), which has found applications in language modeling and machine translation.GANDALF borrows the gating mechanism from GRU (Cho et al., 2014), but modifies it to suit non-temporal, tabular data.

## 3 Gated Adaptive Network for Deep Automated Learning of Features (GANDALF)

In the domain of tabular datasets, achieving state-of-the-art (SOTA) performance often hinges upon two pivotal factors: *Feature Selection* and *Feature Engineering*. GANDALF is meticulously crafted with these core principles at its foundation. The gating mechanism within GANDALF mirrors the resolute stance of Gandalf from *Lord of the Rings* (LOTR). Much like the iconic scene in the Mines of Moria where Gandalf confronts the formidable *Balrog* on a narrow bridge, GANDALF's gating mechanism serves as a powerful barrier against noisy data, echoing Gandalf's legendary command: *"You shall not pass"*.

Suppose we have a training dataset, $\mathcal{D} = (\mathcal{X}, \mathcal{Y}) = \{(x_i, y_i)\}_{i=1}^{S}$, where each $x_i \in \mathbb{R}^d$ is a $d$-dimensional data with a corresponding label $y_i \in \mathcal{Y}$. We assume all the input features are numerical to make notations simpler.

Categorical features are an essential part in tabular data sets and there are to ways we can handle them while using GANDALF. We can pre-process the datasets and encode the categorical features using Target Encoding (Micci-Barreca, 2001) or other similar methods and convert them into a continuous feature. We can also have a learnable Entity Embedding (Guo & Berkhahn, 2016) layer as part of the input which maps categorical features to a fixed length vector, which is learned along with the model.

For classification problems, $\mathcal{Y} = \{1, \ldots, K\}$ and for regression $\mathcal{Y} \in \mathbb{R}$.

There are two main stages in the GANDALF. The input features, $\mathcal{X} \in \mathbb{R}^d$, is first processed by a series of Gated Feature Learning Units (GFLU). The GFLUs learns the best representation of the input features by feature selection and feature interactions. This learned representation is then processed using a simple Multi Layer Perceptron (or even a simple projection) to produce the final prediction.

### 3.1 Gated Feature Learning Units (GFLU)

Gated Feature Learning Unit is an architecture which is inspired by the gating flows in Gated Recurrent Units(Cho et al., 2014). The conventional implementation of the Gated Recurrent Unit (GRU) is specifically designed for processing temporal data, where the same set of features is consistently utilized across multiple time steps, with a same set of weights, while incorporating a memory mechanism that allows the unit to read from and write to it. Mathematically, a GRU is defined as follows (we are ignoring the bias term for brevity):

$$z_t = \sigma(W_z \cdot [H_{t-1}; \mathcal{X}_t]) \tag{1}$$

$$r_t = \sigma(W_r \cdot [H_{t-1}; \mathcal{X}_t]) \tag{2}$$

$$\tilde{H}_t = tanh(W_o \cdot [r_t \cdot H_{t-1}; \mathcal{X}_t]) \tag{3}$$

$$H_t = (1 - z_t) \cdot h_{t-1} + z_t \cdot H_t \tag{4}$$

where, $W_z$, $W_r$, and $W_o$ are learnable weight matrices, $\mathcal{X}_t$ is the input at timestep $t$, and $H_{t-1}$ is the hidden state representation from $t - 1$.

Tabular data often lacks any inherent temporal or sequential nature, devoid of a predefined order among its rows and columns, making the transformation into a sequence unfeasible. To address this challenge, we propose a departure from the temporal aspects traditionally associated with models like the GRU ($t$) and instead introduce a hierarchical formulation involving multiple stages ($n$), each processing the feature input ($\mathcal{X}$).

Unlike its original design tailored for sequence-based operations and consistent transformations across stages, our re-imagined GFLU, tailored for tabular data, diverges from this uniformity in transformations. Specifically, we advocate for distinct weight matrices ($W_n^z, W_n^r, W_n^o$) unique to each stage ($n$), acknowledging that identical transformations at every stage may not be optimal and could even hinder performance.

Moreover, our adaptation of the GRU for tabular data emphasizes the importance of feature selection at each stage. Introducing stage-specific feature masks ($M_n$) allows the learning unit to focus on varying feature subsets, replicating a feature selection process. This re-imagined framework represents a hierarchical Gated Feature Learning Unit ($GFLU$) for tabular data (see Fig. 1). Here, the memory functions as the learned feature representation, iteratively refined by the $GFLU$ at each stage to optimize the data representation for the given task. In contrast to the GRU's objective of learning a unified function within a specific input window, the $GFLU$'s architecture prioritizes the acquisition of diverse functions for individual stages. This facilitates hierarchical stacking, enabling the creation of robust and effective feature representations. We propose stacking these models instead of concatenating because intuitively we see the stacked GFLUs refining the feature representation by choosing the right subset of features stage by stage.

*Fig.* **??** shows a single Gated Feature Learning Unit. We can think of $GFLU$s selecting what information to use from the raw features and using its internal mechanism to learn and unlearn to create the best set of features which is required for the task. The aim of this module is to learn a function $F : \mathbb{R}^d \to \mathbb{R}^{\tilde{d}}$. Although we can use the hidden state dimensions to increase or decrease the dimensions of the learned feature representation ($\tilde{d}$), we have chosen to keep it the same in all our experiments. In subsequent sections, we delve deeper into the intricate details of this architecture.

### 3.1.1 Feature Selection:

We use a learnable mask $\mathbf{M}_n \in \mathbb{R}^d$ and $0 <= \mathbf{M}_n <= 1$ for the soft selection of important features for each stage, $n$, of feature learning in the $GFLU$. The mask is constructed by applying a sparse transformation on a learnable parameter vector ($\mathbf{F}_n \in \mathbb{R}^d$). In 3.2 we talk about how we can initialize this vector using Beta distribution. In order to encourage sparsity, we propose to use the *t-softmax* (Bałazy et al., 2023). They proposed a weighted softmax, as a general form of the softmax.

$$softmax(a_i, w_i) = \frac{w_i \ exp(a_i)}{\sum_{j=1}^{n} w_j \ exp(a_j)} \qquad (5)$$

where $a_i$ is the $i$-th term in the vector over which we are applying the activation, and $w_i$ is the corresponding weight. Under this formulation, if the weight of any term becomes zero ($W_i = 0$), the resulting activated value would also be zero ($softmax(a_i, w_i) = 0$). To make this parameterization easier, they proposed t-softmax, which made all the weights depend on a single parameter, $t > 0$.

$$t\text{-}softmax(a, t) = sofmax(a, w_t),$$
$$\text{where } w_t = ReLU(a_i + t - max(a)) \qquad (6)$$

We can see that all the weights, $w_i$, are positive and at least 1 would be non-zero. This operation is almost as fast as the regular softmax and does not suffer from computational inefficiencies like other sparse activations ($\alpha$-entmax (Peters et al., 2019) and sparsemax (Martins & Astudillo, 2016)) which require a sorting operation. The proposed masking in the GFLU is multiplicative. Formally this feature selection is defined by:

$$M_n = t\text{-}softmax(\mathbf{F}_n, t) \qquad (7)$$

$$\mathcal{X}_n = M_n \odot \mathcal{X} \qquad (8)$$

where $\mathcal{X}_n \in \mathbb{R}^d$ is the input features (after feature selection) and $\mathbf{M}_n = t\text{-}softmax(\mathbf{F}_n)$ and $\odot$ denotes an element-wise multiplication operation. In the *t-softmax*, $t$ can be a learnable parameter or non-learnable.

When it is set to non-learnable, proper initialization of the parameter $t$ is essential. We will go deeper into initialization strategies later. If the parameter, $t$, is learnable, proper initialization is not essential, but helpful.

### 3.1.2 Gating Mechanism

The gating mechanism, which is inspired by Gated Recurrent Units, has a reset and update gate. The feature selection through masks is exclusively employed for the gates, excluding the candidate at stage $n$. This deliberate choice aims to prevent the induced sparsity in the feature masks from impeding gradient flow across the network, and empirical evidence supports its superior performance compared to applying the feature mask to the entire input into the *GFLU*. At any stage, $n$, it takes as an input, $\mathcal{X} \in \mathbb{R}^d$ & $H_{n-1} \in \mathbb{R}^{\tilde{d}}$ as an input and learns a feature representation, $H_n \in \mathbb{R}^{\tilde{d}}$. *Fig.* **??** shows the schematic of a *GFLU*.

At any stage, $n$, the hidden feature representation is a linear interpolation between previous feature representation($H_{n-1}$) and current candidate feature representation($\tilde{H}_n$)

$$H_n = (1 - z_n) \odot H_{n-1} + z_n \odot \tilde{H}_n \tag{9}$$

where $z_n$ is the update gate which decides how much information to use to update its internal feature representation. The update gate is defined as

$$z_n = \sigma(W_n^z \cdot [H_{n-1}; \mathcal{X}_n]) \tag{10}$$

where $[H_{n-1}; \mathcal{X}_n]$ represents a concatenation operation between $H_{n-1}$ and $\mathcal{X}_n$, $\sigma$ is the sigmoid activation function, and $W_n^z$ is a learnable parameter. It's crucial to note that the subscript $n$ emphasizes that the weight varies for each stage of the GFLU.

The candidate feature representation($\tilde{H}_n$) is computed as

$$\tilde{H}_n = tanh(W_n^o \cdot [r_n \odot H_{n-1}; \mathcal{X}]) \tag{11}$$

where $r_n$ is the reset gate which decides how much information to forget from previous feature representation, $W_n^o$ is a learnable parameter, [] represents a concatenation operation, and $\odot$ represents element-wise multiplication. Note that we are using the original feature, $\mathcal{X}$, and not the masked feature, $\mathcal{X}_n$.

The reset gate ($r_n$) is computed similar to the update gate

$$r_n = \sigma(W_n^r \cdot [H_{n-1}; \mathcal{X}_n]) \tag{12}$$

In practice, we can use a single matrix ($W_n^i \in \mathbb{R}^{2d \times 2d}$) to compute the reset and update gates together to save some computation. We can stack any number of such GFLUs to encourage hierarchical learning of features and the feature representation from the last GFLU stage, $\mathcal{H} \in \mathbb{R}^{\tilde{d}}$, is used in the subsequent stages.

### 3.2 Network Architecture and Initialization

GANDALF is a stack of $N$ GFLUs arranged in a sequential manner, each stage, $n$, selecting a subset of features and learning a representation of features, and multiple stages acting in a hierarchical way to build up the optimal representation ($\mathcal{H}$) for the task at hand. Now this representation ($\mathcal{H}$) is fed to a standard $K$ Layer Multi Layer Perceptron with non-linear activations (ReLU), which projects to the output dimension we desire. When $K = 1$, this becomes a simple linear projection layer.

$$\textbf{Classification: } \hat{\mathcal{Y}} = f(\mathcal{H}; W_1, b_1, \ldots, W_K, b_K) \tag{13}$$

$$\textbf{Regression: } \hat{\mathcal{Y}} = T_0 + f(\mathcal{H}; W_1, b_1, \ldots, W_K, b_K) \tag{14}$$

where $W_1, b_1, \ldots, W_K, b_K$ are the weights and biases of the K layer Multi Layer Perceptron and $T_0$ is the average label/target value, which is initialized from training data. $T_0$ makes the model robust to the scale of the label, $\mathcal{Y}$, in case of regression.

Initialization serves as a means to incorporate priors into the architectural framework. To encourage diversity in the hierarchical feature representation learning across different stages, denoted as $n$, we aim for each stage

to have a distinct perspective on the features. Simultaneously, we strive for sparsity in feature selection. Departing from a Gaussian distribution, we opt for initializing $F_n$ with a Beta distribution, renowned for its flexibility with two parameters ($\alpha_n$ and $\beta_n$). To introduce both diversity and sparsity, we stochastically sample $\alpha_n$ and $\beta_n$ from a uniform distribution and leverage the Beta distribution with the sampled parameters to initialize each feature mask.

$$
\begin{aligned}
F_n &\sim \text{Beta}(\alpha_n, \beta_n) \\
\alpha_n &\sim \text{Uniform}(0.5, 10) \\
\beta_n &\sim \text{Uniform}(0.5, 10)
\end{aligned}
\tag{15}
$$

Bałazy et al. Bałazy et al. (2023) had also proposed another variant of the softmax, *r-softmax* where the expected sparsity can be plugged into the activation by using an intuitive parameter they call the sparsity rate, $r$.

$$
\begin{aligned}
r\text{-}softmax(x, r) &= t\text{-}sofmax(x, t_r), \\
\text{where } t &= -quantile(x, r) + max(x)
\end{aligned}
\tag{16}
$$

We do not use this directly in the *GFLU* because the quantile operation is a costly operation. But instead, we use this as a way to initialize the $t$ parameter at the start of training. This is a hyperparameter of the architecture with which we can inject our prior belief about the amount of noise in the input data. Even if the initialization is not spot-on, the network still adjusts itself through training to the right value of $t$.

### 3.3 Interpretability

GANDALF incorporates global feature masks in every stage of its GFLUs, a design choice that inherently enhances interpretability. The learned feature masks from each GFLU stage serve as indicators of the model's reliance on specific features for the task at hand. These individual masks can be aggregated into a comprehensive representation, as illustrated in Eqn: 17. If desired, the aggregated representation can be normalized to limit the range between 0 and 1.

$$
\mathcal{I} = \sum_{n=1}^{N} M_n \qquad \mathcal{I}_i^{norm} = \frac{\mathcal{I}_i}{\sum_{i=1}^{D} \mathcal{I}_i}
\tag{17}
$$

where $N$ is the number of GFLU stages, and $D$ is the number of features in $\mathcal{X}$. This can be seen similar to the feature importance that we get from popular GBDT implementations.

## 4 Experiments and Analysis

In this section, we report the comparison of GANDALF with other models using two publicly available and large-scale benchmarks - *Tabular Benchmark* (Grinsztajn et al., 2022) & *TabSurvey* (Borisov et al., 2022)- along with insights into the hyperparameters and interpretability of GANDALF.We haven't re-run any benchmark, instead took all the results available and evaluaed GANDALF on the same test/validation set. All experiments used PyTorch Tabular (Joseph, 2021) and were run on a single NVIDIA RTX 3060 with 20 cores and 16GB of RAM. The batch size was fixed at 512 for all experiments except *Higgs* dataset for the *TabSurvey* benchmark. Higgs being a very large dataset (7.7 Million rows), we chose a higher batch size (4096) faster training, mixed precision training to manage limited GPU memory and sub-epoch level check-pointing to track progress.

### 4.1 Comparison with Public Benchmarks

The selection of datasets for benchmarking is non-standard in the tabular domain. Nevertheless, open benchmarks now provide standardized datasets and evaluation metrics. Leveraging these benchmarks ensures transparency and comparability in our evaluation with other studies. Our evaluation protocol aligns with the benchmarks' methodology; while we refrain from re-running models already tested by benchmarks, we assess GANDALF in the same test/validation splits. For Bayesian Optimization (BO), we employ the Optuna

library (Akiba et al., 2019) with a Tree Parzen Estimator (Bergstra et al., 2011) for *100* trials. Adhering to open benchmarks, we maintain consistency in train-test splits, pre-processing, cross-validation folds, and metrics. Results are reported as the average score on the test split for Tabular Benchmark and on the validation split for *TabSurvey*, following the benchmarks' conventions. Standard deviation of scores is reported for experiments with more than one fold.

Additionally, we utilize Thop (Zhu, 2013) to compute the number of parameters and MACs (Multiply-Accumulate Operations)[1] for the best-performing deep learning models and present the findings.

When aggregating across different datasets, we have used the average distance to the minimum (ADTM) metric (Wistuba et al., 2015; Grinsztajn et al., 2022). It is a simple affine transformation which scales the performances between 0 and 1. $ADTM(s \in S) = \frac{s-min(S)}{max(S)-min(S)}$, where $S$ is all the scores of best models in a dataset. In *TabSurvey*, regression is measured using mean squared error, which is a lower-is-better metric and we adapted ADTM to be $ADTM(s \in S) = \frac{s-max(S)}{max(S)-min(S)}$ to be consistent with the interpretation.

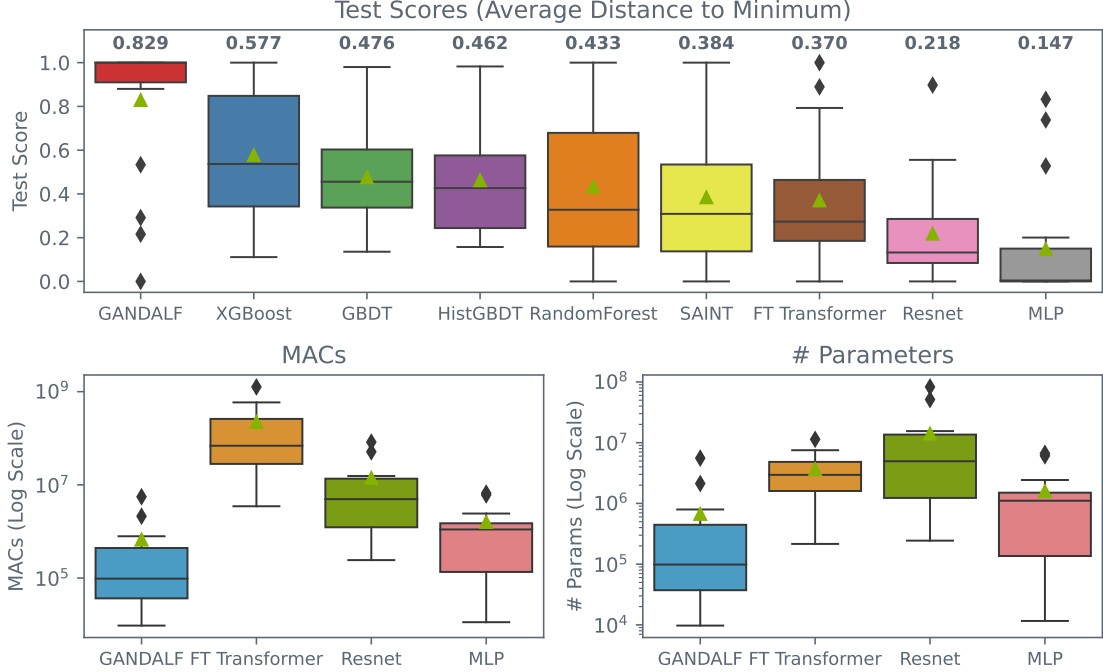

Figure 3: Benchmarking on *Tabular Benchmark* Grinsztajn et al. (2022). The box plot of the normalized test scores across datasets show that GANDALF consistently achieve the best scores. The box plots of the number of parameters and MACs(Multiply-Accumulate Operations) show that GANDALF achieves the high performance with higher parameter efficiency and computational efficiency. Individual scores for all datasets are in the Appendix A

### 4.1.1 Tabular Benchmark

*Tabular Benchmark* (Grinsztajn et al., 2022) used 45 datasets from varied domains, carefully curated to capture the variety of tabular datasets and made publicly available through OpenML (Bischl et al., 2017). They ran separate benchmarking for numerical and categorical classification/regression, medium and large sized datasets, with and without target transforms, data transforms, etc., accounting for more than 20,000 compute hours. To limit our computational burden and make the evaluation on datasets resembling real-world problems, we have selected a subset of 18 datasets and benchmarks from this according to the following rules:

---

[1]According to https://github.com/sovrasov/flops-counter.pytorch/issues/16#issuecomment-518585837, 1 MAC is roughly equal to 2 FLOPs

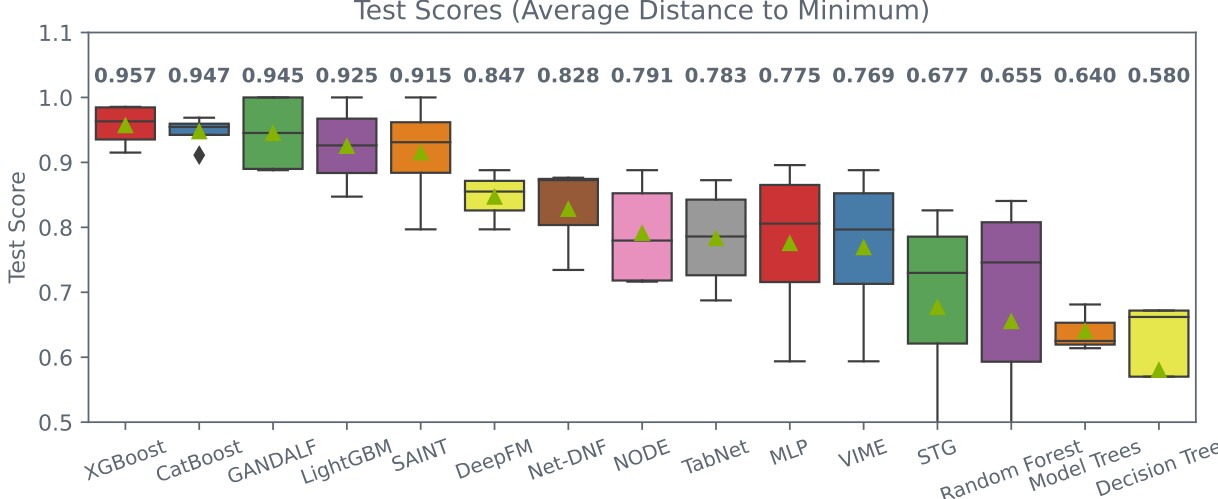

Figure 4: Benchmarking on *TabSurvey* Borisov et al. (2022). The box plot of the normalized test scores across 4 datasets shows GANDALF being at par with GBDTs and better than other DL models.Individual scores for all datasets are in the Appendix A

1. Only medium sized datasets, i.e. maximum size of all the datasets is 10000

2. No target or data transfomations

3. Only datasets with more than 20 features

4. If a dataset was used for both numerical and categorical benchmarks, we chose the categorical benchmark.

Fig. 3 shows the box plots of the normalized test scores across datasets and the number of parameters and MACs for deep learning models. Detailed results available in the Appendix. These results will not match the ones in Grinsztajn et al. (2022) because the results in the paper is aggregated over the entire set of datasets whereas we have used the same train and test splits but of a subset of the full benchmark and used the detailed results they released as a csv for comparison.

## 4.2 Tabsurvey Benchmark

We also used a smaller benchmark (Borisov et al., 2022), with just 5 datasets, to ensure larger coverage on existing tabular models. The benchmark covers 20 models, 12 from Deep Learning. We excluded *Heloc* dataset because the processed dataset (train and test split) wasn't available publicly. For all the other datasets, we have used the same train and test/validation splits and tuned and trained GANDALF. We used the same definition of ADTM to aggregate scores across the four datasets and the results are plotted in Fig. 4. Detailed results available in the Appendix.

## 4.3 Results

In this section, we present the results of our experiments, showcasing the superior performance of GANDALF across multiple dimensions.

- Benchmarking on *Tabular Benchmark* Grinsztajn et al. (2022):
    - *GANDALF* consistently achieves the best scores across a diverse set of 18 datasets, showcasing its robust performance on medium-sized datasets with varied characteristics.

- The box plots in Fig. 3 emphasize *GANDALF*'s superior accuracy and its ability to maintain high parameter and computational efficiency.
        - Detailed results, including standard deviations for experiments with multiple folds, are available in the Appendix.
    - Benchmarking on *TabSurvey* (Borisov et al., 2022):
        - *GANDALF* performs at par with Gradient Boosted Decision Trees (GBDTs) and outperforms other Deep Learning models on the *Tabsurvey* benchmark, as depicted in Fig. 4.
        - The box plots highlight *GANDALF*'s competitive test scores across four datasets, reinforcing its effectiveness in diverse tabular scenarios.
    - The clear superior performance relative to a standard Multilayer Perceptron (MLP) underscores the clear advantage of incorporating Gated Feature Learning Units (GFLUs) as a representation learning layer over a plain MLP.
    - *GANDALF* outperforms various alternatives, including those utilizing attention mechanisms like *FTTransformer* and *SAINT*. This underscores the effectiveness of *GANDALF*'s gating mechanism in feature selection and engineering.

In summary, our experiments demonstrate that GANDALF excels not only in accuracy but also in computational and parameter efficiency, making it a compelling choice for tabular data analysis.

## 4.4 Ablation Study

Ablation Study to evaluate design choices was carried out on a subset of 7 datasets. We evaluated three main design decisions in the architecture:

1. Use of GFLUs vs plain MLPs

2. Use of sparse activations like *entmax*, *t-softmax*, *sparsemax* vs regular *softmax*

3. Use of beta distribution to initialize feature masks vs regular random initialization using normal distribution

All three design decisions shows definite bump in test scores on most datasets (Figure 11 in Appendix A). It underlines the fact that the inductive bias encoded in the GFLUs are beneficial for tabular data.

## 4.5 Hyperparameter Study

GANDALF, with its highly expressive architecture, requires only three crucial hyperparameters—*# of GFLU Stages, Dropout in each GFLU, and Init Sparsity*. In contrast, other deep learning approaches like NODE, SAINT, Tabnet, and even popular GBDT packages often involve managing a more extensive set of hyperparameters. For all our experiments, we kept the dimensions of the Multi Layer Perceptron constant with two hidden layers of 32 and 16 units respectively with a ReLU activation, and set the sparsity parameter as learnable. Hyperparameter search was done on the below search space for *100* trials using BO, with first 10 trials as pure random exploration.

$$\text{GFLU Stages} \in \{2, 3, \ldots, 30\}$$
$$\text{GFLU Dropout} \sim \text{Uniform}(0.0, 0.5)$$
$$\text{Init Sparsity} \sim \text{Uniform}(0.0, 0.9)$$
$$\text{Weight Decay} \in \{10^{-3}, 10^{-4}, 10^{-5}, 10^{-6}, 10^{-7}, 10^{-8}\}$$
$$\text{Learning Rate} \in \{10^{-3}, 10^{-4}, 10^{-5}, 10^{-6}\}$$

We should notice that the only architecture specific hyperparameter that was tuned was the # of GFLU stages and GFLU Dropout. To study how fast the tuning converges, we ran simulated scenarios where

identified the best performing model among all models if we had run only $T$ trials, where $T \in 1, 2, \ldots, 20$. Out of 18 datasets in *Tabular Benchmark*, GANDALF was the best performing model in 13 of them as early as 5 trials (where we are still in the random exploration phase). Fig. 8 in Appendix shows the detailed plot. We conducted an in-depth analysis of hyperparameter tuning across 18 datasets to identify crucial parameters

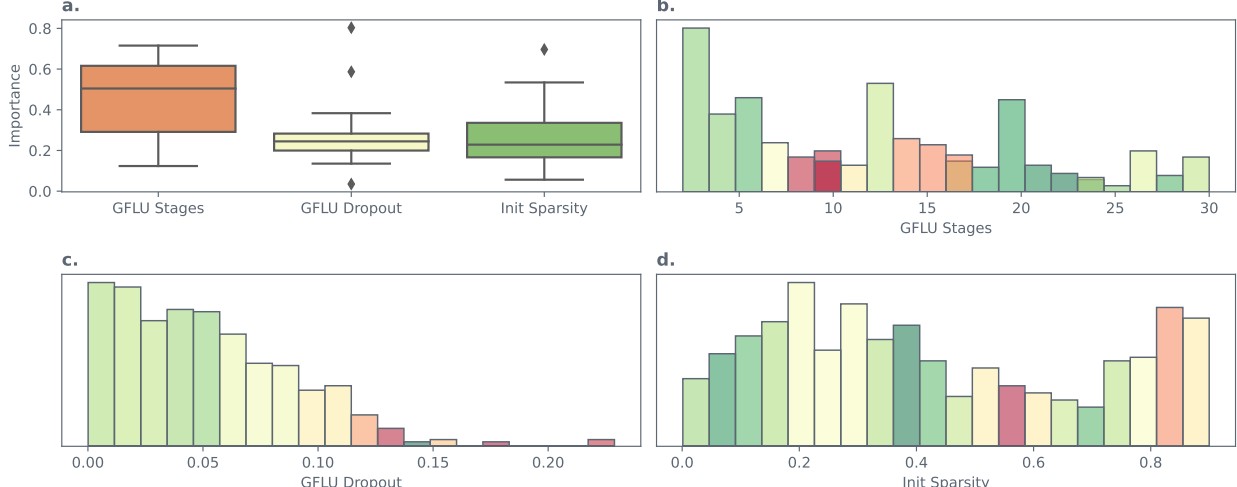

Figure 5: Hyperparameter Study. a. The box plot of hyperparameter importances across 18 datasets. b - d. Histograms of hyperparameters where each bin is colored according to the average test scores in that bin. Green is higher and Red is lower.

and establish effective default ranges for tuning. We used the Optuna implementation of FANOVA (Hutter et al., 2014) to assess the relative importance of each hyperparameter. To isolate the influence of Learning Rate and Weight Decay, we focused exclusively on trials utilizing the optimal values for these parameters. We also plotted the histogram of trial values for *# of GFLU Stages, GFLU Dropout*, and *Init Sparsity* and colored each bin based on the average normalized test score to give an indication of the possible ranges of the hyperparameters. *Fig. 5* shows the corresponding plots.The key observations are:

1. Even though the sparsity parameter is learnable, giving it a good starting point helps quite a bit. But we can see that a sparsity between 0.0 and 0.5 is a good range to search.

2. *GFLU Dropout* has its sweet spot between 0 and 0.05. This can also be because the *GFLU Dropout* is applied to each GFLU stage and hence compounding the effect.

3. *# of GFLU Stages* doesn't show any particular pattern, but is having the highest impact on performance. This tells us tuning the *# of GFLU Stages* is essential.

## 4.6 Interpretability

We carried out two experiments to analyze the interpretability of GANDALF. For both experiments, we only used datasets from the *Tabular Benchmark* which has no categorical features. This was done only to make the analysis simpler. In case of categorical features, we will just need to average the $\mathcal{I}$s associated with the corresponding embeddings.

### 4.6.1 Analysis using Synthetic Dataset

We used a few synthetic datasets with only a few informative features and ran GANDALF to figure out if the model is able to isolate the useful variables from the noise. We used a dataset with 8 features, 4 of which are informative and the rest noise. For all experiments, we ran with 6 *GFLU Stages* and *Init Sparsity* of 0.1 (lower Init Sparsity leads to clearer Interpretability because each stage will focus on a handful of features). Fig

6 shows the results and we can see that the gating mechanism in GANDALF was able to isolate informative features, as evidenced by the dark green and dark blue boxes.

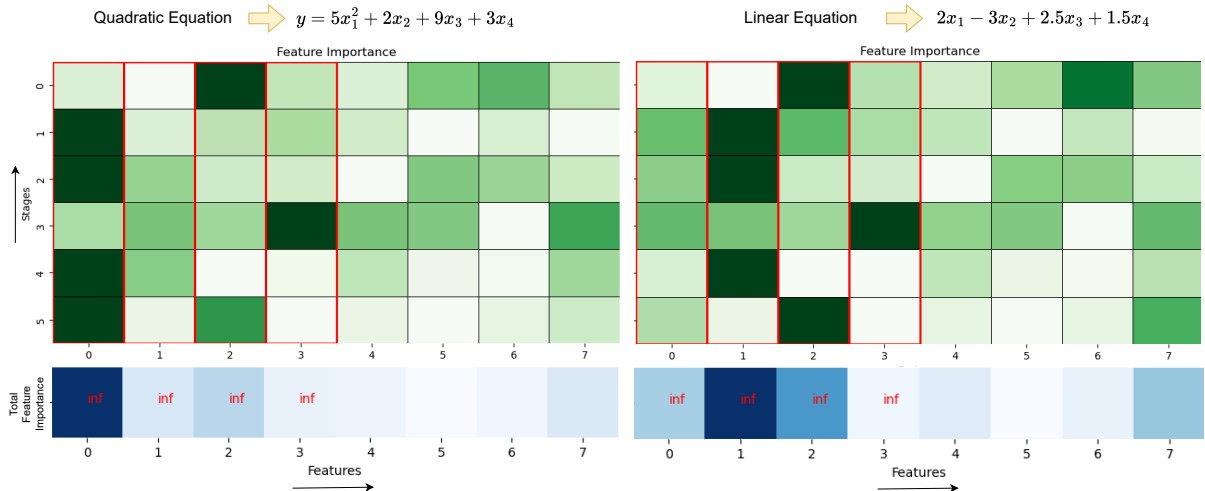

Figure 6: Stage-wise Feature Importance (in greens) and total feature importance (in blues) for two synthetic datasets - a linear equation and a quadratic equation. First 4 features are informative and have been outlined with red.

### 4.6.2 Ranking Metrics of the Feature Importance

Evaluating feature attributions or importance is not a trivial task. Since there is no ground truth for checking the quality of interpretability, we decided to use GradientSHAP (Lundberg & Lee, 2017) and DeepLIFT (Shrikumar et al., 2017) as a measure of ground truth. We formulated the assessment as a recommendation problem where the ground truth is the top 25 features according to GradientSHAP and DeepLIFT. We compare the ranked feature importance from best performing configurations of GANDALF and XGBoost (as a baseline) using *Normalized Discounted Cumulative Gain@K(NDCG@K)* (Järvelin & Kekäläinen, 2002) at different values of $K \in 1, 5, 10, 15, 25, 50$. *Fig. 7a* and *Fig. 7b* plots show the results. We can see that the NDCG scores of the *GANDALF* feature importance are consistently better than the baseline, *XGBoost*, when compared against GradientSHAP and DeepLIFT ground truths. This shows that the GANDALF feature importance is at least as good as the popularly used alternative for GBDTs.

### 4.6.3 Feature Ablation Study

Another quantifiable aspect of an explanation is it's *fidelity*. *Fidelity* measures how well an explanation represents the processing performed by a model on its inputs to produce the output. If a feature importance ranking is true, then if we perturb the information in each feature from most relevant features first (MoRF) (Tomsett et al., 2020), the performance of the model should decrease rapidly. And if we do it in the reverse order, Least Relevant First (LeRF), we should not expect huge drops in performance. Borisov et al. (2022) also uses this technique to assess the fidelity of the feature attributions. But they removed the feature and retrained the model without the feature. We chose to keep the model the same and replaced each feature with Gaussian noise as a perturbation. Retraining the model without the feature can also change the way the model is configured and may not measure the fidelity of the model we are interested in anymore. *Fig. 7b* and *Fig. 7c* plots show the *MoRF* and *LeRF* curves for GANDALF feature importance. We can see that as expected, the performance drops sharply when we remove most relevant features first (MoRF) and it falls less rapidly when we remove least relevant features first(LeRF). This further shows the effectiveness of the gating mechanism in *GANDALF* and the fidelity of the extracted feature importances.

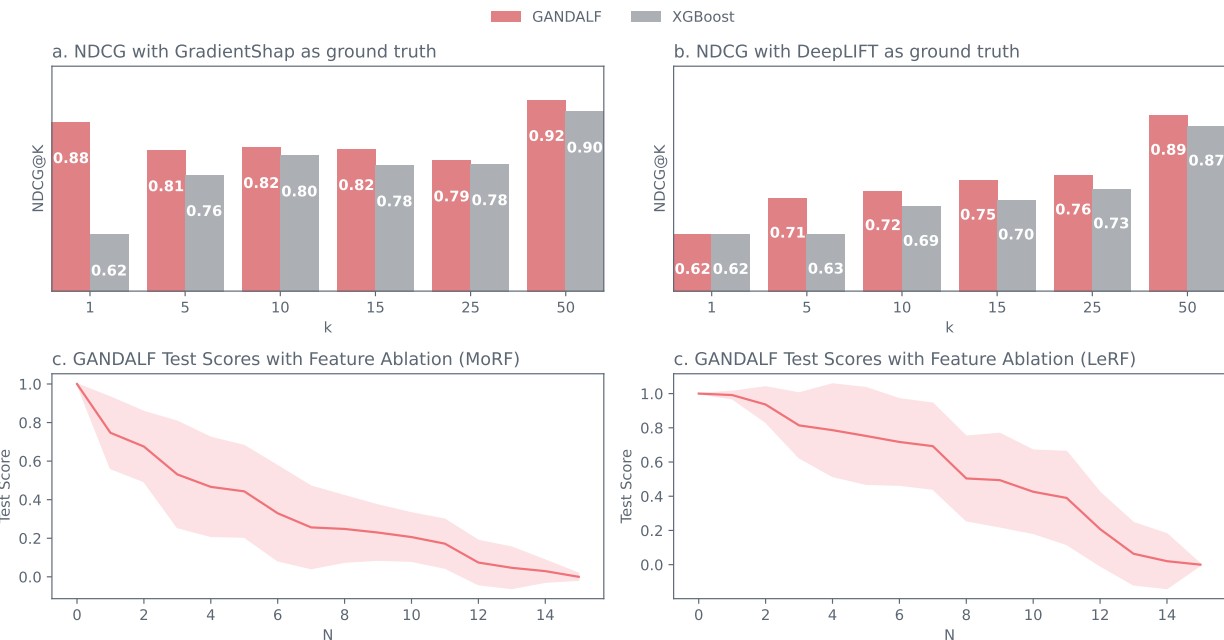

Figure 7: Interpretability Study. a - b. The NDCG@K scores for GANDALF and XGBoost Feature importance against GradientSHAP and DeepLIFT. c - d. Median normalized test scores in feature ablation trials for Most Relevant First (MORF) and Least Relevant First(LORF) strategies.

# 5   Conclusion

We introduce GANDALF (Gated Adaptive Network for Deep Automated Learning of Features), a pioneering deep learning architecture designed for tabular data. At its core is the Gated Feature Learning Unit (GFLU), a modular element designed for tabular data processing. Our extensive experiments across diverse benchmarks not only position GANDALF as a competitive alternative to popular Gradient Boosted Decision Tree (GBDT) models but also showcase its superiority over other prevalent deep learning methods tailored for tabular data. Remarkably, GANDALF achieves this with fewer parameters and computations, emphasizing its computational efficiency.

However, it's crucial to acknowledge that GANDALF, like any model, has its limitations. One significant constraint lies in its inherent sequential processing across stages, which precludes parallel processing. Despite this restriction, the sequential computation remains viable owing to its lightweight computations and parameter efficiency.

The fundamental strength of GANDALF lies in its prowess for feature selection and learning. Consequently, its advantages might be more pronounced in datasets with a higher feature count or where the relationship between features and target outcomes isn't straightforward. Nevertheless, empirical observations across various real-world datasets affirm that GANDALF consistently offers substantial advantages over standard Multi-Layered Perceptrons (MLPs), maintaining a comparable computational footprint.

Beyond performance metrics, GANDALF offers inherent explainability, making it a valuable asset for practitioners seeking insights into model decisions. The deliberate minimization of architecture-specific hyperparameters enhances accessibility, simplifying the training process for users. In essence, GANDALF encapsulates a fusion of superior performance, interpretability, computational efficiency, and accessibility, presenting a promising leap forward in the realm of tabular deep learning. The source code has been made available on `https://anonymous.4open.science/r/gandalf-43BD` under the MIT License.

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

# A  Additional Tables and Figures

Table 1: Summary of datasets from Tabular Benchmark(Grinsztajn et al., 2022) used in the experiments.

| Benchmark | Dataset | Type | # Folds | # Features | Train Size | Test Size |
|---|---|---|---|---|---|---|
| Categorical Classification | albert | Classification | 1 | 31 | 10000 | 33777 |
| Categorical Classification | covertype | Classification | 1 | 54 | 10000 | 50000 |
| Categorical Classification | default-credit-card | Classification | 3 | 21 | 9290 | 2788 |
| Categorical Classification | eye_movements | Classification | 3 | 23 | 5325 | 1599 |
| Categorical Classification | road-safety | Classification | 1 | 32 | 10000 | 50000 |
| Categorical Regression | Allstate | Regression | 1 | 124 | 10000 | 50000 |
| Categorical Regression | Mercedes | Regression | 5 | 359 | 2946 | 885 |
| Categorical Regression | topo_2_1 | Regression | 3 | 255 | 6219 | 1867 |
| Numeric Classification | Bioresponse | Classification | 5 | 419 | 2403 | 722 |
| Numeric Classification | Higgs | Classification | 1 | 24 | 10000 | 50000 |
| Numeric Classification | MiniBooNE | Classification | 1 | 50 | 10000 | 44099 |
| Numeric Classification | heloc | Classification | 3 | 22 | 7000 | 2100 |
| Numeric Classification | jannis | Classification | 1 | 54 | 10000 | 33306 |
| Numeric Classification | pol | Classification | 3 | 26 | 7057 | 2118 |
| Numeric Regression | Ailerons | Regression | 3 | 33 | 9625 | 2888 |
| Numeric Regression | cpu_act | Regression | 3 | 21 | 5734 | 1721 |
| Numeric Regression | superconduct | Regression | 1 | 79 | 10000 | 7885 |
| Numeric Regression | yprop_4_1 | Regression | 3 | 42 | 6219 | 1867 |

Table 2: Summary of datasets from Tab Survey(Borisov et al., 2022) used in the experiments.

| Dataset | Type | Categorical Features | # Folds | # Features | Train Size | Test Size |
|---|---|---|---|---|---|---|
| adult-income | Classification | True | 1 | 14 | 22792 | 9769 |
| Higgs | Classification | True | 1 | 28 | 7700000 | 3300000 |
| covertype | Classification | False | 1 | 54 | 406708 | 174304 |
| california-housing | Regression | False | 2 | 8 | 14448 | 6192 |

Table 3: Test scores on Tabular Benchmark(Grinsztajn et al., 2022). For classification Accuracy and for regression $R^2$ score is reported. The best performing model is highlighted in bold.

| Dataset | GANDALF | XGBoost | HistGBDT | GBDT | RandomForest | FT Transformer | Resnet | SAINT | MLP |
|---|---|---|---|---|---|---|---|---|---|
| albert | **68.53** | 65.70 | 65.78 | 65.76 | 65.53 | 65.63 | 65.23 | 65.52 | 65.32 |
| covertype | **93.22** | 86.58 | 84.98 | 85.43 | 85.86 | 85.93 | 83.96 | 85.33 | 83.32 |
| default-credit-card | **72.80** | 72.08 | 72.00 | 72.09 | 72.13 | 71.90 | 71.40 | 71.90 | 71.41 |
| eye_movements | **67.89** | 66.85 | 64.23 | 64.67 | 66.21 | 60.00 | 59.97 | 60.56 | 60.54 |
| road-safety | **80.88** | 76.89 | 76.42 | 76.31 | 76.13 | 77.09 | 76.09 | 76.69 | 75.59 |
| Allstate | **59.11** | 53.65 | 52.74 | 53.04 | 49.70 | 52.01 | 51.41 | 52.60 | 51.58 |
| Mercedes | **59.84** | 57.87 | 57.89 | 57.76 | 57.81 | 56.63 | 57.29 | 56.45 | 55.91 |
| topo_2_1 | 4.85 | 6.94 | 7.31 | 5.34 | **7.36** | 5.33 | 5.06 | 6.05 | 4.15 |
| Bioresponse | **88.16** | 79.31 | - | 78.59 | 79.86 | 75.82 | 77.06 | 76.87 | 76.70 |
| Higgs | **75.23** | 71.37 | - | 71.04 | 70.93 | 70.61 | 69.48 | 70.82 | 68.97 |
| MiniBooNE | 92.42 | 93.72 | - | 93.44 | 92.72 | 93.51 | 93.67 | **93.81** | 93.45 |
| heloc | 72.26 | 72.16 | - | 72.35 | 72.10 | **72.67** | 72.41 | 72.37 | 72.40 |
| jannis | **79.35** | 78.03 | - | 77.47 | 77.30 | 76.83 | 75.37 | 77.12 | 74.57 |
| pol | **98.98** | 98.33 | - | 98.16 | 98.24 | 98.50 | 95.22 | 98.21 | 94.70 |
| Ailerons | **86.46** | 83.66 | 84.49 | 83.43 | - | 73.28 | 71.85 | 70.10 | 83.72 |
| cpu_act | 98.29 | **98.62** | 98.35 | 98.61 | - | 98.48 | 98.26 | 98.51 | 97.91 |
| superconduct | **92.57** | 91.06 | 90.16 | 90.32 | - | 89.03 | 89.53 | - | 89.65 |
| yprop_4_1 | 8.53 | 8.29 | - | 5.57 | **9.39** | 5.35 | 4.28 | 5.95 | 2.23 |

---

[2]We binned Latitude and Longitude features and considered as Categorical Feature

Table 4: Standard deviation of test scores on Tabular Benchmark Grinsztajn et al. (2022) (Only Datasets where k-Fold Validation was carried out). For classification Accuracy and for regression $R^2$ score is reported.

| Dataset | GANDALF | XGBoost | HistGBDT | GBDT | RandomForest | FT Transformer | Resnet | SAINT | MLP |
|---|---|---|---|---|---|---|---|---|---|
| default-credit-card | 5.01E-03 | 6.30E-03 | 6.59E-03 | 7.43E-03 | 4.31E-03 | 3.98E-03 | 1.08E-02 | 5.69E-03 | 5.76E-03 |
| eye_movements | 9.21E-03 | 3.57E-03 | 7.36E-03 | 4.09E-03 | 5.22E-03 | 1.29E-02 | 6.68E-03 | 7.08E-03 | 1.16E-02 |
| Mercedes | 1.59E-02 | 8.41E-01 | 8.46E-01 | 8.42E-01 | 8.20E-01 | 8.35E-01 | 8.38E-01 | 8.71E-01 | 7.91E-01 |
| topo_2_1 | 2.32E-02 | 4.21E-04 | 4.19E-04 | 4.42E-04 | 4.75E-04 | 4.67E-04 | 4.99E-04 | 5.35E-04 | 4.23E-04 |
| Bioresponse | 2.06E-02 | 1.74E-02 | - | 1.26E-02 | 1.32E-02 | 1.56E-02 | 1.00E-02 | 1.53E-02 | 1.32E-02 |
| heloc | 3.13E-03 | 1.06E-02 | - | 8.31E-03 | 1.31E-02 | 1.42E-02 | 1.26E-02 | 9.08E-03 | 1.02E-02 |
| pol | 1.01E-03 | 1.24E-03 | - | 2.34E-03 | 1.98E-03 | 2.36E-03 | 6.41E-03 | 3.91E-03 | 5.42E-03 |
| Ailerons | 1.12E-03 | 4.03E-06 | 4.58E-06 | 5.12E-06 | - | 1.14E-05 | 1.14E-05 | 2.54E-05 | 1.86E-06 |
| cpu_act | 2.75E-04 | 6.73E-02 | 2.13E-01 | 3.96E-02 | - | 2.56E-02 | 2.50E-02 | 6.83E-02 | 9.01E-02 |
| yprop_4_1 | 2.95E-02 | 4.75E-04 | - | 5.40E-04 | 4.79E-04 | 3.96E-04 | 5.26E-04 | 4.37E-04 | 3.89E-04 |

Table 5: Test scores on TabSurvey(Borisov et al., 2022). For classification Accuracy and for regression Mean Squared Error score is reported. The best performing model is highlighted in bold and Standard deviation is reported as a ± range.

| Method | Adult
Acc↑ | HIGGS
Acc↑ | Covertype
Acc↑ | Cal. Housing$^2$
MSE↓ |
|---|---|---|---|---|
| XGBoost | 87.3 | 77.6 | 97.3 | 0.206±0.005 |
| CatBoost | 87.2 | 77.5 | 96.4 | 0.196±0.004 |
| GANDALF | 86.7 | 76.9 | **97.7** | **0.16±0.006** |
| LightGBM | **87.4** | 77.1 | 93.5 | 0.195±0.005 |
| SAINT | 86.1 | **79.8** | 96.3 | 0.226±0.004 |
| DeepFM | 86.1 | 76.9 | - | 0.260±0.006 |
| Net-DNF | 85.7 | 76.6 | 94.2 | - |
| NODE | 85.6 | 76.9 | 89.9 | 0.276±0.005 |
| TabNet | 85.4 | 76.5 | 93.1 | 0.346±0.007 |
| MLP | 84.8 | 77.1 | 91.0 | 0.263±0.008 |
| VIME | 84.8 | 76.9 | 90.9 | 0.275±0.007 |
| STG | 85.4 | 73.9 | 81.8 | 0.285±0.006 |
| Random Forest | 86.1 | 71.9 | 78.1 | 0.272±0.006 |
| Model Trees | 85.0 | 69.8 | - | 0.385±0.019 |
| Decision Tree | 85.3 | 71.3 | 79.1 | 0.404±0.007 |
| TabTransformer | 85.2 | 73.8 | 76.5 | 0.451±0.014 |
| DeepGBM | 84.6 | 74.5 | - | 0.856±0.065 |
| RLN | 81.0 | 71.8 | 77.2 | 0.348±0.013 |
| KNN | 83.2 | 62.3 | 70.2 | 0.421±0.009 |
| Linear Model | 82.5 | 64.1 | 72.4 | 0.528±0.008 |
| NAM | 83.4 | 53.9 | - | 0.725±0.022 |

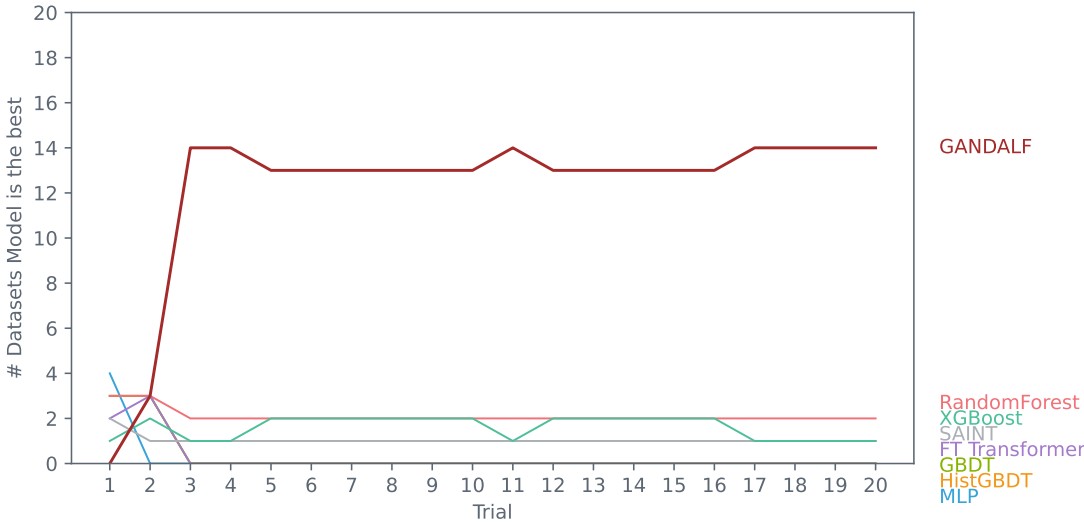

Figure 8: Hyperparameter Tuning Scenarios: If the random iterations were only N trials, how many datasets do a model become the best. We can see that GANDALF quickly takes the lead in 13 of 18 datasets in 5 trials (which is still in the random exploration stage of the hyperparameter tuning.

Table 6: Tuned Hyperparameters for each dataset in Tabular Benchmark (Grinsztajn et al., 2022).

| Dataset | GFLU Stages | Init Sparsity | GFLU Dropout | Learning Rate | Weight Decay |
|---|---|---|---|---|---|
| default-credit-card | 2 | 0.870769 | 0.021064 | 0.001 | 0.001 |
| heloc | 13 | 0.291414 | 0.072046 | 0.001 | 0.0001 |
| eye_movements | 2 | 0.802989 | 0.011162 | 0.001 | 0.001 |
| Higgs | 10 | 0.717149 | 0.087058 | 0.001 | 1e-05 |
| pol | 3 | 0.181439 | 0.103777 | 0.001 | 1e-07 |
| albert | 18 | 0.007152 | 0.014137 | 0.001 | 1e-07 |
| road-safety | 19 | 0.001146 | 0.063497 | 0.001 | 1e-05 |
| MiniBooNE | 14 | 0.115137 | 0.002363 | 0.001 | 0.001 |
| covertype | 7 | 0.067178 | 0.056627 | 0.001 | 0.0001 |
| jannis | 2 | 0.309363 | 0.023717 | 0.001 | 1e-06 |
| Bioresponse | 2 | 0.103818 | 0.000808 | 0.0001 | 1e-05 |
| cpu_act | 21 | 0.529018 | 0.000493 | 0.001 | 1e-07 |
| Ailerons | 2 | 0.880892 | 0.003350 | 0.001 | 1e-07 |
| yprop_4_1 | 9 | 0.193272 | 0.006030 | 0.001 | 1e-05 |
| superconduct | 21 | 0.151234 | 0.000635 | 0.001 | 1e-05 |
| Allstate | 27 | 0.609144 | 0.100092 | 0.0001 | 1e-05 |
| topo_2_1 | 14 | 0.817548 | 0.057329 | 0.001 | 0.0001 |
| Mercedes | 30 | 0.031092 | 0.003284 | 0.0001 | 0.0001 |

Table 7: Tuned Hyperparameters for each dataset in TabSurvey (Borisov et al., 2022).

| Dataset | GFLU Stages | Init Sparsity | GFLU Dropout | Learning Rate | Weight Decay |
|---|---|---|---|---|---|
| adult-income | 20 | 0.242888 | 0.051258 | 0.001 | 0.001 |
| Higgs | 19 | 0.030949 | 0.085262 | 0.001 | 1e-05 |
| covertype | 18 | 0.36086 | 0.00226 | 0.001 | 0.0001 |
| california-housing | 14 | 0.121434 | 0.012372 | 0.001 | 1e-07 |

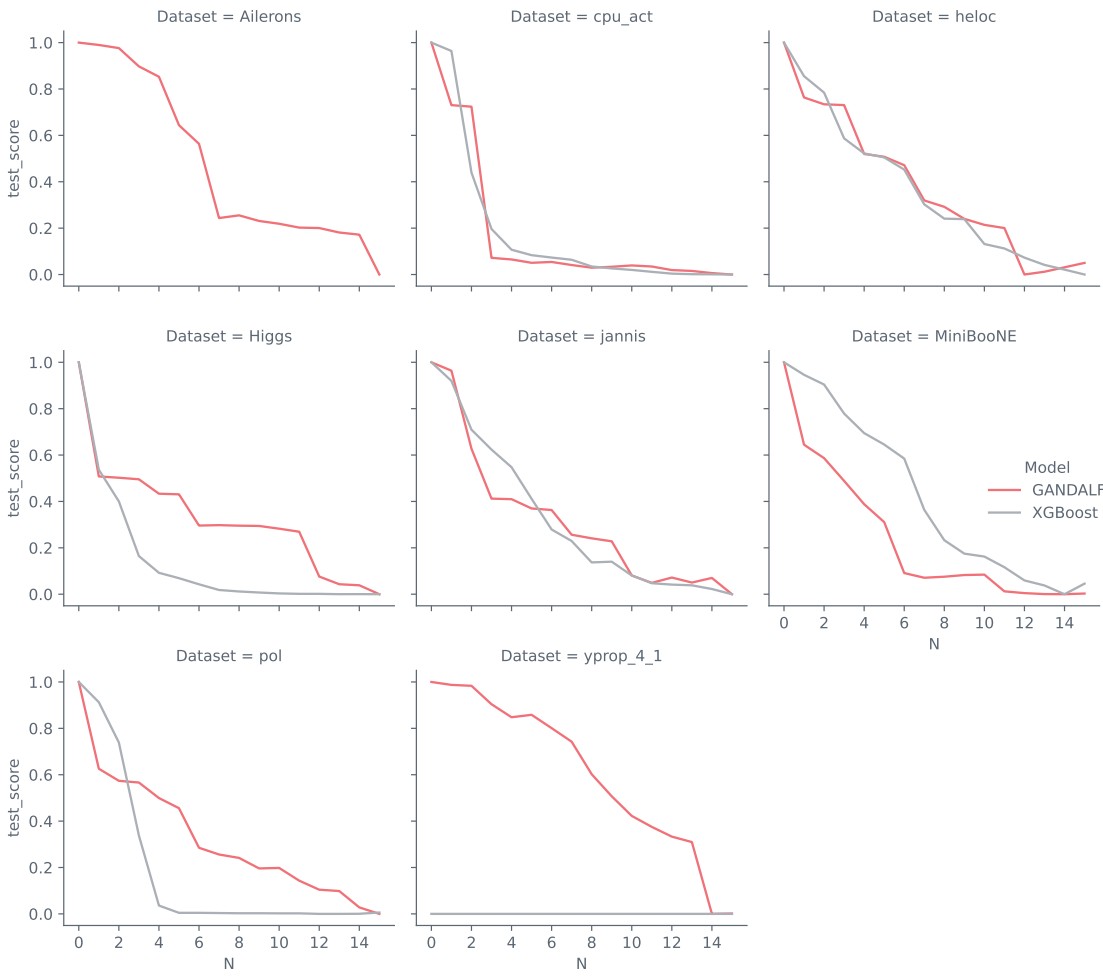

Figure 9: Feature Ablation (MoRF): GANDALF vs XGBoost for all datasets. We can see that the Test Scores drop drastically as we take out most relevant features, comparable to the XGBoost Feature importance.

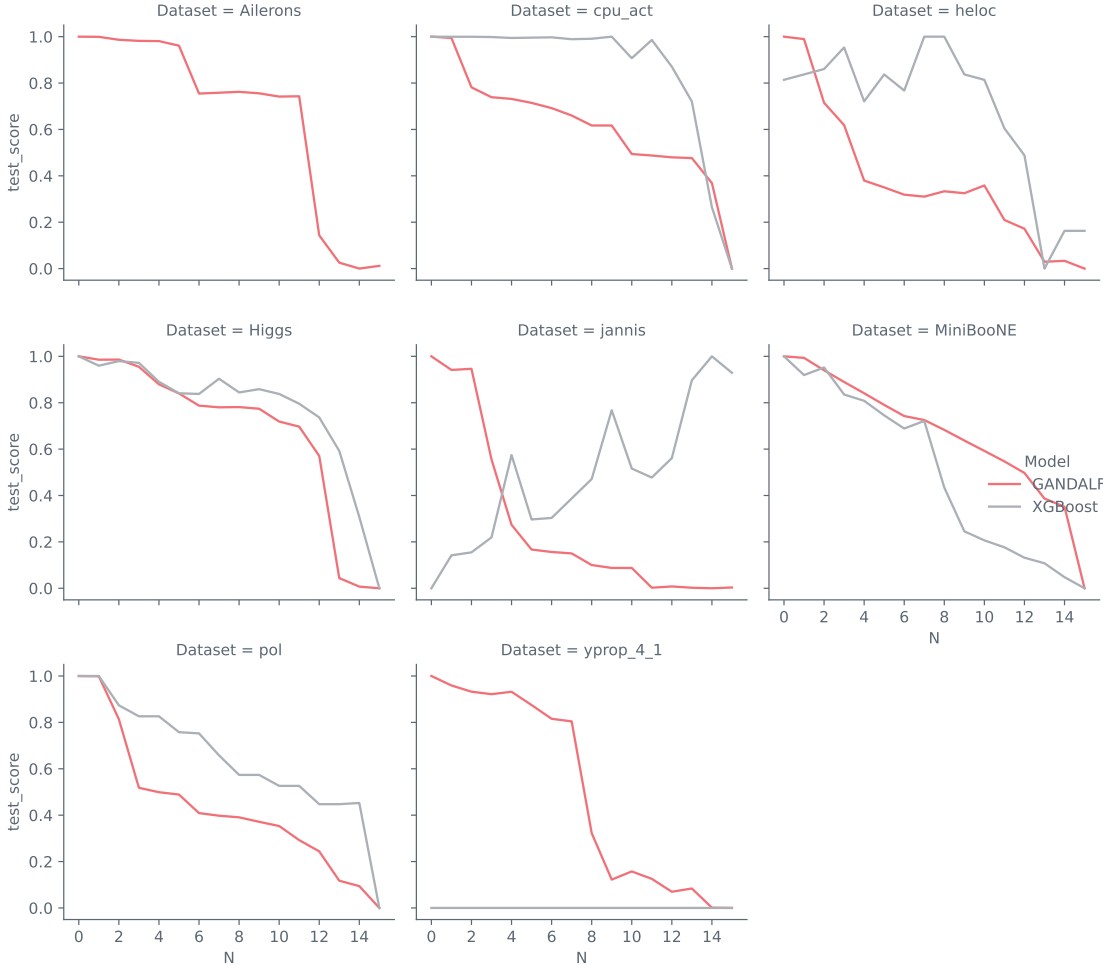

Figure 10: Feature Ablation (LeRF): GANDALF vs XGBoost for all datasets. We can see that the Test Scores do not drop as drastically as in MoRF as we take out least relevant features. *yprop_4_1* is a hard dataset and XGBoost test scores were too low to make an appearance in the chart.

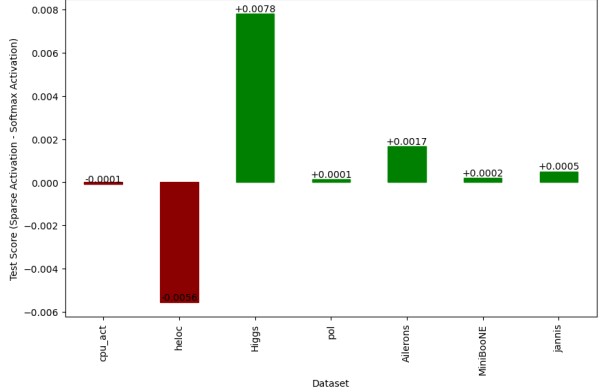

(a) The difference in test scores using sparse activations vs softmax

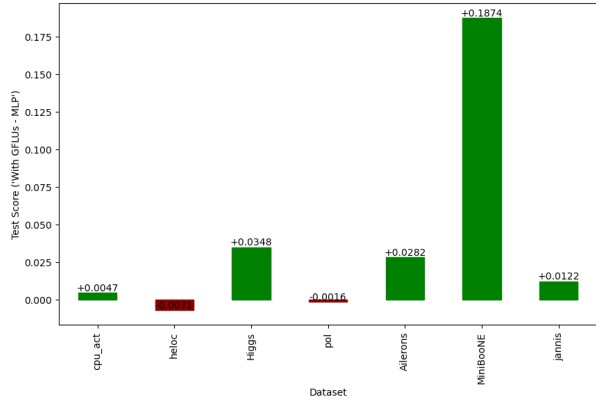

(b) The difference in test scores using GFLU layers vs simple MLPs

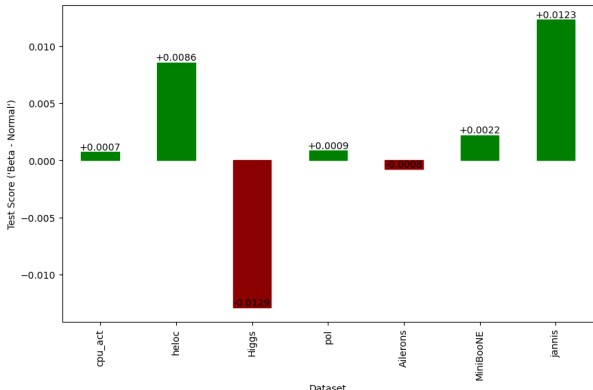

(c) The difference in test scores between using Beta distribution to initialize feature mask vs normal distribution

Figure 11: Ablation Study to evaluate design choices was carried out on a subset of 7 datasets. All three design decisions shows definite bump in test scores on most datasets. It underlines the fact that the inductive bias encoded in the GFLUs are beneficial for tabular data.

