# OpenReview forum: "GANDALF: Gated Adaptive Network for Deep Automated Learning of Features for Tabular Data"
_TMLR — Rejected by TMLR_

### Review · Reviewer_8egK · 2024-07-03

**Summary Of Contributions:**

The authors present an approach for feature extraction for tabular data based on a, to the best of my knowledge, novel application of neural network architecture component inspired by Gated Recurrent Units (GRUs). The proposed method is tested on two public tabular benchmarks [Grinsztajn et al, 2022](https://arxiv.org/abs/2207.08815) and [Borisov et al., 2021](https://arxiv.org/pdf/2110.01889).

Experimental evaluations suggest that the proposed method has on par ([Borisov et al., 2021](https://arxiv.org/pdf/2110.01889)) or substantially better performance ([Grinsztajn et al, 2022](https://arxiv.org/abs/2207.08815)) compared to other methods.

Comparisons of the feature relevance computed by the proposed method appears to mimic the behaviour of other established feature relevance or explainable artificial intelligence (XAI) approaches, such as Shapley Value approximations.

Improving the state of the art in tabular prediction tasks is a relevant contribution. Especially as mounting empirical evidence seems to suggest that tree based methods appear to outperform neural network (NN) based methods across the board in the very same data sets as used in this study.

The authors of ([Grinsztajn et al, 2022](https://arxiv.org/abs/2207.08815)) investigate the superiority of trees over NNs and find that one factor could be the inductive bias of tree based methods and their robustness towards noisy features. This robustness can be attributed in part to the boosting components of many popular tree based algorithms.

The results of the present study could be interpreted as bridging this gap between NNs and tree based methods: the way the gating mechanism is used (at first it seemd counterintuitive to me to use a method commonly used for sequences, GRU, for tabular data) is reminiscent of boosting strategies with every mask in figure 1, right, representing a weak learner.

So in principle this could make sense, that using something like boosting in a NN (?) improves NNs -- but the improvements seen in the ([Grinsztajn et al, 2022](https://arxiv.org/abs/2207.08815)) benchmark appears to be somewhat at odds with the results in that original study: [Grinsztajn et al, 2022](https://arxiv.org/abs/2207.08815) find that NNs perform consistently worse than tree based methods, sometimes by a large margin. And the present study claims that GANDALF achieves substantial improvements over all methods also tested in ([Grinsztajn et al, 2022](https://arxiv.org/abs/2207.08815)).

(I'm focussing on that study as the proposed method appears to perform very well in that benchmark, while in the other benchmark, the proposed method is not better than prior work.)

While it is possible that the relative metrics are really correct and so much better than previous approaches, it seems that the performances for tree based methods in ([Grinsztajn et al, 2022](https://arxiv.org/abs/2207.08815)) are better than in the present study. Especially it seems counterintuitive that the values reported in the appendix, table 3, are so different from the results reported in ([Grinsztajn et al, 2022](https://arxiv.org/abs/2207.08815)): in that study, the NN methods ResNet and FT Transformer perform consistently significantly worse than all tree based methods. In the present study, it is exactly those methods that appear to beat GANDALF (and all tree based methods).

I might be missing something but it seems that hyperparameter optimization was only done exhaustively for the proposed GANDALF method? And all other results were taken from existing benchmarks? If the baseline methods were not HPO'd extensively this would explain the inferior performance of tree based methods, but that would be an unfair comparison.

So I'm assuming that the authors used the existing results of the published benchmarks, but then it would be helpful to provide some more details on the test splits (were they the same?) and maybe also provide the absolute metrics. And ideally comment on the discrepancies with previous results.

The authors argue that in order to limit the computational burden, writing that '*Only datasets with more than 20 features*' were selected - wouldn't it be ok to use data sets with less than 20 features, I mean just considering computational complexity?

On the interpretability / XAI part: I'm not sure I understand why the interpretability aspects should be compared with other existing methods, however popular they might be. Comparing the quality of interpretability methods with actual ground truth could be done with synthetic data for instance - that would allow for better control over the conditions when the feature selection works and when it doesn't.

**Audience:**

Yes

**Claims And Evidence:**

No

**Requested Changes:**

For a better assessment of the empirical pros and cons of the proposed method, it would be great if the authors could:

* add some more detail on the baseline method results -- were they fitted by the authors themselves? If so, it would be good to highlight how the HPO was done to make sure results are comparable. If not, it would be good to mention how the test splits were done.
* it would be interesting to see some ablation studies to assess which aspects of the proposed architectural component is key to this substantial improvement of the performance
* for interpretability results, it would be helpful to compare that with the feature selection robustness as in previous studies - for instance fig. 5 in Grinsztajn et al, 2022.

**Strengths And Weaknesses:**

## Strengths
* Novel application of GRU component for tabular data
* extensive empirical evaluation
* potentially interesting explanation for previously inferior performance of neural networks, compared to tree based methods, on tabular data


## Weaknesses
* Results seem a bit at odds with previous empirical evidence
* some details on the HPO (if applicable) would be helpful
* XAI experiments use Shapley approximations as proxy - ground truth could be obtained from synthetic data for instance

---

> ### Author Response · Authors · 2024-07-19
>
> Thank you for taking time to review the paper in such detail. Let us address the major concerns one by one:
>
> •	The apparent discrepancy between the results in the paper and Grinsztajn et al, 2022 is because of the fact that Grinsztajn et al, 2022 reports the aggregate figures in the paper. We have taken a subset of the large benchmark with the specified conditions and evaluated our model in that subset. Grinsztajn et al, 2022 have released the entire list of random search that they have done in a csv and have also released the code and train-val-test splits that they have used for each dataset. We have taken that subset, same train-val-test splits and evaluated GANDALF on it.  We will add a few lines in the revised paper to emphasis this point.
>
> •	FT Transformer and ResNets do not perform better than GANDALF or GBDTs in the subset of Grinsztajn et al, 2022 benchmark. In fact, in our subset as well, both these models perform worse than Random Forest (Fig 3 Top chart). The bottom two charts in Fig 3 are for the MACs and number of parameters and shows that the best GANDALF model has lesser # parameters and MACs than the best Ft Transformer and ResNets (These charts have lower the better property)
>
> •	Hyperparameter Tuning was only done for GANDALF as the benchmarks have already tuned all the competing methods extensively. Grinsztajn et al, 2022 have released Train-Val-Test splits. We have used the val split for the hyperparameter tuning and evaluated the best model on Test. TabSurvey Borisov et al. (2022 have done their tuning using just a train and val split and reported performance on the validation dataset and we have followed the same to align the reporting. For GANDALF, we have used Bayesian Optimization with 100 trials. Details on the grid are in Section 4.4 and the tuned hyperparameter for each dataset is in the Appendix(Table 6 and 7)
>
> •	Selecting datasets > 20 features was to make it similar to real world datasets, whereas the other conditions were to reduce computational burden.
>
> •	Using synthetic data to evaluate the interpretability was a good suggestion and we have added a section with those results in the revised paper.
>
> •	The scope for an ablation study was limited as the key components in the architecture are minimal – The gating mechanism, and sparse activations. We have carried out ablation study on 7 of the datasets in Grinsztajn et al, 2022 to check the effect of sparse activations vs softmax, and using GFLUs + MLPs, and just MLPs. This chart will be included in the Appendix (Fig 11) in the revised paper. Out intuition is that the Gating mechanism and stage by stage learning is what drives most of the performance.
>
> •	The feature selection robustness (Fig 5 from Grinsztajn et al, 2022) is already part of the current study, but in a slightly different perspective. Fig 7 c and 7d shows very similar experiments – Taking the Least relevant feature according to the feature importance from the data set and retraining to see how performance drops and vice versa. And the synthetic dataset study also shows the feature selection robustness in the face of noise features.
>
> We will update the revised paper in a few days once the third review is also available

---

> > ### Author Response · Authors · 2024-08-03
> >
> > We have uploaded the revised paper with the changes requested.

---

### Review · Reviewer_AdKS · 2024-07-08

**Summary Of Contributions:**

The paper introduces GANDALF, a new deep learning architecture for tabular data. This approach utilizes a modified GRU unit that eliminates a temporal parameter. The authors demonstrate GANDALF's strong performance on two challenging tabular data benchmarks and show that the selection maps from the GFLU layer can be used for interpretability purposes.

**Audience:**

Yes

**Claims And Evidence:**

Yes

**Requested Changes:**

## Recommendations

Mostly addressing the weaknesses above:

- Expand the list of baselines to include recent models
- Conduct a thorough ablation study to justify design choices
- Include informative plots similar to those in TabForestPFN [1] and "Why do tree-based models still outperform deep learning on typical tabular data?" by Léo Grinsztajn et al.
- The idea of the gates is not novel, and has been used for the feature importance analysis and for inference tasks [2, 3, 4], perhaps authors can provide indicate the similarities and research for more references in this direction
 - Additionally, clarify how GANDALF handles categorical data, as this is a crucial aspect of tabular data processing.


References:

[1] Breejen, Felix den, et al. "Why In-Context Learning Transformers are Tabular Data Classifiers." arXiv preprint arXiv:2405.13396 (2024).

[2] Yamada, Yutaro, et al. "Feature selection using stochastic gates." ICML, pp. 10648-10659. PMLR, 2020.

[3] Borisov, Vadim, et al. "Cancelout: A layer for feature selection in deep neural networks." ICANN 2019, pp. 72-83. Springer, 2019.

[4] Li, Yifeng, et al. "Deep feature selection: theory and application to identify enhancers and promoters." Journal of Computational Biology 23.5 (2016): 322-336.

**Strengths And Weaknesses:**

## Strengths
- New deep learning architecture that performs well across various tabular datasets and offers the interpretability property
- Robust benchmarking methodology using challenging and open-source frameworks
- Convincing performance results

## Weaknesses
- Lack of discussion on handling categorical data, a significant challenge in tabular data processing
- Some of the recent baselines are missing, TabPFN, TabForestPFN, DeepTLF, and so forth
- Insufficient justification for design decisions (e.g., selection t-softmax and other architectural choices) due to the absence of an ablation study

## Minor Issues
- Oversized Figures 1 and 2; consider reducing their dimensions + Uninformative captions for these figures

---

> ### Author Response · Authors · 2024-07-19
> **Responses**
>
> Thank you for spending your valuable time for reviewing the paper and recognizing the importance of the work. Let us try to address a few of your concerns:
> •	Categorical Data can be either pre-encoded using Target Encoding to convert them into numerical features or it can be encoded with the model using a learnable embedding layer. There is a line in Section 3 introduction which talks about this. We didn’t elaborate because it is a common practice in tabular data, but we will add a bit more focus in the revised paper.
>
> •	For competing baselines, we were relying on publicly available data benchmarks for two benefits: 1. The published results are something which can be replicated easily. 2. We need not spend out compute and time to evaluate competing models, but instead just our model. That is why we were restricted to the models that these public benchmarks have included.
>
> •	Also, the models like TabPFN uses in-context learning and may not be able to scale to some of the large datasets that are used for the benchmark. DeepTLF is a hybrid model which uses Decision Trees as a preprocessing layer and a NN as a subsequent predictor. We feel “DeepGBM: A deep learning framework distilled by GBDT for online prediction tasks” has a much more detailed study of possible ways of hybridization and this is already part of the model list in the TabSurvey benchmark. GANDALF is shown to beat DeepGBM (in the detailed results in the appendix. Deep GBM wasn’t included in the fina overall comparison because Tabsurvey benchmark didn’t have DeepGBM results for Covertype dataset). Also, through TabSurvey we have also compared GANDALF to newer models like Net-DNF, Neaural Additive Models, VIME, etc. We have also showed that GANDALF is much better than the other feature selection based technique (Y. Yamada, O. Lindenbaum, S. Negahban, and Y. Kluger, “Feature selection using stochastic gates”) in the Tabsurvey benchmark.
>
> •	The scope for an ablation study was limited as the key components in the architecture are minimal – The gating mechanism, and sparse activations. We have carried out ablation study on 7 of the datasets in Grinsztajn et al, 2022 to check the effect of sparse activations vs softmax, and using GFLUs + MLPs, and just MLPs. We will include this in the revised paper in Appendix (Fig 11). Out intuition is that the Gating mechanism and stage by stage learning is what drives most of the performance.
>
> •	We did some research in the feature selection direction you pointed out and thank you for those references. If we look at our GFLU, there are two components – the Feature Masking and the Gating mechanism. The feature selection using masking is not novel and have been used in other prior tabular models like Tabnet as well as the other references you shared. Our masking strategy is most closest to the one in TabNet, but instead of a sparsemax, we proposed to move to t-softmax (this is mainly because t-softmax is much faster than sparsemax or entmax and t-softmax also lets us inject priors into the desired sparsity). We will include this direction in the literature review.
>
> •	The informative plots in TabForestPFN and Grinsztajn et al, 2022 communicate two main things – How different models, in general, are performing as compared to each other, How much tuning is necessary fir different models to gain required performance. We believe we have included both that information about GANDALF. Fig 3 and Fig 4 shows GANDALF performance when compared to other models. And Fig 8 in appendix shows how quickly GANDALF gains performance while tuning (in 13 out of 18 datasets, GANDALF became the best model in as little as 5 trials in hyperparameter tuning.
>
> We will update the revised paper in a few days once the third review is also available

---

> > ### Author Response · Authors · 2024-08-03
> >
> > We have uploaded the revised paper with the changes requested.

---

### Review · Reviewer_4VJb · 2024-07-29

**Summary Of Contributions:**

The authors of the paper present a framework towards feature selection and feature engineering.
To do so, firstly they introduce Gated Feature Learning Units (GFLU) an alternative architecture to Gated Recurrent Unit (GRU, Cho et al., 2014).
The main difference between GRU and GFLU is that in GFLU, the weight matrices are updated at each time step.

The feature selection task is based on t-softmax (Bałazy et al., 2023), an alternative function to softmax in which a smooth transition between softmax and binary one-hot vectors is attempted. The authors employ the t-softmax function in order to generate a masking matrix M_n through which they perform the feature selection.

The authors suggest a feature representation method which is designed with a gating mechanism at which an update and a reset gate are are employed at each step. The former is designed aiming to decide how much information will be used during the internal feature representation update and the latter is designed aiming to decide how much information will be forgotten during each internal feature representation update.

The authors then proceed with presenting experimental results.

**Audience:**

Yes

**Claims And Evidence:**

No

**Requested Changes:**

The authors would be kindly asked to:

1) explain in detail how the M_n (page 5) is specified, whether its output is binary or not and, if not, how is the masking operation performed;
2) consider tabular data properties, like imbalanced columns of categorical data, and explain how their method can tackle them;
3) consider additional metrics, more precisely F1 score and ROC, at the classification tasks;
4) Comment why their method does not seem to perform well in feature ablation. More precisely, at Figures 8 and 9:
a) compare their method to all 8 methods presented at Figure 2, instead of doing so only with XGBoost;
b) explain their method similar performance to XGBoost and underperformance in datasets cpu_act, heloc, higgs, jannis and MinBooNE;

**Strengths And Weaknesses:**

The authors of the paper extend the idea of GRU towards non-temporal data. They attempt so through a gating mechanism, namely GFLU, that is tailored so that the corresponding weight matrices of each gate and feature representation are updated at each step.

The authors claim that their methods is designed for tabular data although they do not seem to have considered properties of categorical/discrete data like imbalanced categorical columns or sparse one hot vectors (referring to the feature value, not to the feature weight).
The authors also employ a masking matrix M in R^(t by d) , where d is the feature dimension and t the number of steps but its output is not clearly specified, as it is not detailed whether the assumptions of Bałazy et al., 2023 are suffied.

---

> ### Author Response · Authors · 2024-08-03
>
> Thank you for taking time to review the paper. Let us address the major concerns one by one:
>
> 1.	M_n is the activated vector of F_n, which in turn is a free learnable parameter. F_n is of the same dimension as the number of features and is initialized using a beta distribution. All this is mentioned in the paper but is fragmented and we will make our best effort to make it clearer in the revised paper. The free learnable parameter F_n is passed through the t-softmax layer to get sparse activations, which is the mask, M_n. And Balazy et al. 2023 only have an assumption that the weights of the weighted softmax be positive. But the way they have constructed t-softmax and its variant, r-softmax ensures that there is no negative weights no matter what the data it is passed to. The weights used in t-softmax is of this form: w^i_t = ReLU(x_i+t-max(x)). We can see that there is a ReLu they apply to ensure any negatives are truncated to zeros and it is also how they are getting the sparsity from.
>
> 2.	Categorical Data can be either pre-encoded using Target Encoding to convert them into numerical features or it can be encoded with the model using a learnable embedding layer. There is a line in Section 3 introduction which talks about this. We didn’t elaborate because it is a common practice in tabular data, but we will add a bit more focus in the revised paper and make it clearer. And proposed model isn’t specific to imbalanced data and it’s a common problem for any algorithm/model. Typical strategies like sample weights, over and under sampling, etc. would work with proposed model as well.
>
> 3.	We have used standard open benchmarks to compare our model with maximum number of competing models on same data/test splits with minimum compute. Therefore, we have aligned to the metrics that were used in such benchmarks. But these benchmarks have been chosen in such a way that the there is no imbalance in the class labels. This makes the F1 score/ROC comparable to plain accuracy.
>
> 4.	Feature ablation tests are a way to see if the feature importance holds relevance to the predictive performance. And we are comparing it to XGBoost to just have a benchmark on a dataset. But when compared to tree-based models which does hard feature selection, the soft feature selection of GANDALF will not be as clean as XGBoost. If we draw parallels between the Trees in XGBoost and GFLUs in GANDALF, the Trees can completely ignore features with its hard feature selection, but GFLUs use soft feature selection. So even if GANDALF identifies the least relevant feature, perturbing it would still impact some  of the GFLUs, affecting its performance. Therefore, GANDALF showing slightly slower decline in MoRF and slightly higher decline in LeRF than XGBoost is expected. But as long as the expected shape is followed, we can be confident that the feature importance is indicative of predictive performance.
>
>         4a.	Comparing with other methods: Again, following the same logic that we included XGBoost curves as a rough benchmark to show the expected shape of the curve. While it is possible to repeat the experiment with other models, we used XGBoost as a representative from established SOTA group (GBDTs) which also gives feature importance. Other methods like SAINT or MLPs do not have any feature importance. We also measure “goodness” of feature importance from GANDALF by comparing it to model agnostic interpretability techniques like GradientSHAP and DeepLIFT. There also, we included XGBoost as a representative of the GBDT class of models to act as a benchmark.
>
>         4b.	We wouldn’t term it as under performance, but in datasets like jannis, cpu_act, heloc, we can see that the LeRF curves dip too fast for GANDALF. There can be many reasons for it from dataset specific (almost all features are indicative of predictive performance), learned model specific (model may have learned with lesser sparsity, using combinations of features rather than individual features for each GFLUs, or may not have learned a good model at all), and so on. cpu_act and heloc were difficult datasets for GANDALF, as evidenced by lower scores (Table 3) and maybe GANDALF wasn’t able to learn the data well. jannis is a dataset where GANDALF excelled and the sharp drop in jannis after a few least relevant features may indicate that the model was relying on all the features to make accurate predictions.
>
> We have uploaded the revised paper

---

### Decision · Action_Editor_3YsW · 2024-09-10

**Recommendation:** Reject

**Comment:**

Reviewer 8egK notes: "The authors did a great job at adressing all points raised in the reviews. Great effort was put into additional experiments and clarification."

The submission meets the bar in terms of Audience but not in terms of Claims and Evidence (see above), and as a result it is not quite ready to be published in TMLR. That being said, the submission has several merits and I could see it becoming significantly stronger after incorporating Reviewer 8egK's suggestions on experimental design. I would encourage the authors to do so and resubmit their work.

**Audience:**

All reviewers agree that the submission is of interest to the TMLR audience, as evidenced by the answer to the Audience question in their official recommendation. Reviewer 8egK also notes: "I believe this work is a valuable contribution as it proposes a novel neural network architecture component that appears to improve the so far not so great predictive performance of the neural network model class on tabular data."

**Claims And Evidence:**

Most reviewers still have reservations on the Claims and Evidence acceptance criterion that center around the evidence presented to support the claim that GANDALF is competitive with GBDT and superior to other prevalent deep learning methods tailored for tabular data.

The main point of contention is the experimental design decision made by the submission to exclude datasets in Grinsztajn et al. (2022) with 20 features or less, prompting reviewers to wonder whether GANDALF's competitive performance is restricted to that particular dataset regime. Reviewers are not convinced by the authors' justification in terms of "real-lifeness" and computational requirements. In their official recommendation, Reviewer 8egK writes:

"It would be totally ok for this method to have a slightly lower performance in those experiments/data sets that were excluded. But excluding those experiments raises doubts about the quality of the work. And I think that the work presented is great, so why weakening the argument by excluding data sets from the comparison?

"I'd strongly suggest to use exactly the same data as the Grinsztajn et al, 2022 paper -- this would strengthen the paper substantially."

**Resubmission Of Major Revision:**

The authors may consider submitting a major revision at a later time.